# Epidemiological patterns of motorcycle-related injuries in Cameroon: A comparative analysis of motorcycle users and pedestrians

Elvis A. Tanue[1,2]*, Odette D. Kibu[1,3], Betrand A. Tambe[1], Nicholas Tendongfor[1,2], Alan Hubbard[2,3,4], Sandra I. McCoy[2], Nahyeni Bassah[2], Rasheedat Oke[2,3,5], Isaac Obeng-Gyasi[5], Arole Darwin Touko[3], Fanny Nadia Dissak-Delon[3], S. Ariane Christie[5], Georges Nguefack-Tsague[2,3,6], Dickson S. Nsagha[1], Catherine Juillard[2,3,5], Alain Chichom-Mefire[2,3]

1 Department of Public Health and Hygiene, Faculty of Health Sciences, University of Buea, Buea, Cameroon, 2 Sustainable Trauma Research Education and Mentorship (STREaM) Project, Faculty of Health Sciences, University of Buea, Buea, Cameroon, 3 Data Science Center for the Study of Surgery, Injury, and Equity in Africa (D-SINE Africa), University of Buea, Buea, Cameroon, 4 Division of Epidemiology, School of Public Health, University of California Berkeley, Berkeley, California, United States of America, 5 Program for the Advancement of Surgical Equity, Department of Surgery, University of California Los Angeles, Los Angeles, California, United States of America, 6 Department of Public Health, Faculty of Medicine and Biomedical Sciences, University of Yaoundé 1, Yaounde, Cameroon

* elvis.tanue.a@gmail.com

## Abstract

Motorcycles are a common mode of transport in major Cameroonian cities, contributing to a rising burden of injuries among both users and pedestrians. These groups differ in exposure, mechanisms, and vulnerability, yet both bear a disproportionately high injury burden. However, comparative data on their epidemiological patterns and outcomes remain scarce. To support targeted prevention policies, we analysed trauma registry data to describe demographic, crash, injury, clinical, and outcome characteristics across both populations. This was a retrospective analysis of the Cameroon Trauma Registry (CTR), which collects information on injured patients presenting to 10 hospitals across seven of the 10 regions of Cameroon. Patients presenting with motorcycle-related injuries between June 1st 2022 and May 31st 2023 were assessed for demographic, crash, injury, clinical patterns of care and outcomes variables and compared using χ² or Fisher's exact tests for categorical data. The analysis was done using R version 4.2.1. A total of 2,757 motorcycle-related injury patients were included from the CTR database, including 2,339 (84.8%) motorcycle users and 418 (15.2%) pedestrians. Motorcycle users were mostly aged 15–34 years (59.1%) and males (83.0%), while pedestrians were frequently aged ≥60 years (23.4%) and females (37.8%). Helmet use among motorcycle users was low (3.0%). Alcohol involvement was more frequent among users (14.2%) than pedestrians (7.4%, p = 0.001). Most injuries occurred during work for users (33.1%) and during leisure for pedestrians (77.5%, p < 0.001). Severe multi-region injuries (abbreviated

**Data availability statement:** All relevant data are within the paper and its Supporting information files.

**Funding:** The funds were awarded to CJ and AC as principal investigators by the National Institute of health with grant number U54TW012087. https://www.nih.gov/. The sponsors did not participate in writing the manuscript. The funders had no role in study design, data collection and analysis, decision to publish, or preparation of the manuscript.

**Competing interests:** The authors have declared that no competing interests exist.

injury severity ≥3) were more common in users (21.9%) than pedestrians (16.1%, p = 0.014). Hospital admissions were high in both motorcycle users (60.2%) and pedestrians (58.4%); 6.2% required intensive care, and 2.2% underwent immediate surgery. Functional outcomes were similar: 44.8% had minor and 19.9% had major disability at discharge; 3.6% died during hospitalization. Motorcycle-related injuries in Cameroon disproportionately affect young male motorcycle users, with low helmet use, higher rates of alcohol use and severe trauma, together with older female pedestrians. Despite differing profiles, both motorcycle users and pedestrians face high disability and hospitalization rates. Targeted safety strategies are urgently needed to address these overlapping and distinct risks.

## Background

Injuries remain a neglected global health challenge, responsible for over 5 million deaths annually and nearly 10% of the global burden of disease, with transport injuries ranking among the top contributors to disability and mortality [1,2]. Low- and middle-income countries (LMICs) carry over 90% of this burden, and road traffic injuries (RTIs) are the leading cause of death among individuals aged 5–29 years [3,4]. Despite global declines in transport injury rates, LMICs continue to experience disproportionate morbidity, mortality, and economic losses, with annual costs equivalent to 3% of their gross domestic product [3,5].

Cameroon exemplifies this crisis where it records approximately 16,500 road crashes and nearly 6,000 deaths annually, translating into 1,443 disability-adjusted life years (DALYs) lost per 100,000 population, one of the highest rates in the world [6]. The Douala–Yaoundé highway, the nation's busiest, has a fatality rate 35 times higher than comparable roads in Europe and North America [7]. Beyond the human toll, RTIs in Cameroon impose economic losses exceeding 100 billion CFA francs (approximately 165 million USD) annually [8].

Motorcycles have become the dominant mode of urban transport due to affordability and accessibility, but their rapid adoption has coincided with an alarming rise in severe injuries. In Ngaoundéré, motorcycles accounted for 60% of RTI victims, predominantly involving head and limb trauma [9]. In Douala, prospective data showed that most motorcycle crashes involved collisions with cars or other motorcycles, occurring mainly at night or weekends, with only 5.7% of motorcycle riders wearing helmets at the time of the crash [10]. A study in Kumba found that the majority of RTIs presenting to hospitals were motorcycle-related, with head and limb injuries predominating [11]. Commercial motorcycle riders who are often untrained, unlicensed, and operating in unsafe conditions constitute the largest group of victims [10,12].

However, pedestrians also represent a major proportion of RTI casualties and, together with motorcyclists, constitute the most vulnerable road users. Hospital-based surveillance in Yaoundé found that pedestrians comprised 34% and motorcyclists

29% of RTI admissions, both sustaining severe injuries [13]. Similarly, other studies across sub-Saharan Africa identify these two groups as vulnerable road users with frequent and serious injuries [14]. Despite this, most Cameroonian studies have focused on either group in isolation, motorcyclists [9–12] or general trauma populations [15] limiting the ability to identify differential risk factors and outcomes.

Risk factors for RTIs consistently reported include poor road infrastructure, speeding, overloading, alcohol consumption, and non-use of protective gear [8–12]. Yet, helmet wearing remains dismally low nationwide: only 2.7% of riders reported helmet use in recent Cameroon Trauma Registry (CTR) data from 10 hospitals across seven regions [16]. Prehospital care is another weak point, with only 5% of patients receiving any scene care, though such interventions are associated with improved survival [16].

Encouragingly, recent efforts in trauma surveillance and quality improvement demonstrate feasibility and impact. Implementation of trauma registries has improved data completeness and characterization of RTIs [17], while trauma quality improvement bundles have strengthened resuscitation practices, including vital-sign collection and airway management [17]. Nevertheless, existing studies remain fragmented, often hospital-based, and region-specific, limiting their generalizability.

Given that motorcycle users and pedestrians together constitute the majority of RTI victims in Cameroon, each with distinct crash mechanisms, injury profiles, and outcomes, a comparative analysis is warranted. Such evidence is essential to guide targeted preventive strategies, improve enforcement of safety regulations, and inform investment in prehospital and hospital trauma systems tailored to the needs of these vulnerable groups.

## Methods

### Ethics statement

The CTR obtained ethical approval from the University of Buea Institutional Review Board (2021/1506–07/UB/SG/IRB/ FHS, approved on 30th September 2021), the Cameroon National Ethics Committee (N˚2018/09/1444/CE/CNERSH/ SP, approved on 25th March 2022), the University of California, Los Angeles Institutional Review Board (#19–000086, approved on 13th June 2019), and administrative authorization from the Cameroon Ministry of Public Health (D30-923/L/ MINSANTE/SG/ DROS, approved on 8th August 2022). Written informed consent was obtained from patients or guardians where appropriate. Data were de-identified prior to analysis.

### Study design and setting

We conducted a retrospective analysis of prospectively collected data from the Cameroon Trauma Registry (CTR), a hospital-based surveillance system that records consecutive injury patients presenting to sentinel facilities. This study examined data from June 1, 2022, through May 31, 2023 (12 months). The data were accessed for research purposes in 15/08/2023. The CTR operates at 10 hospitals spanning seven of Cameroon's ten regions: Laquintinie Hospital of Douala, Limbe Regional Hospital, Pouma Catholic Hospital, Edea Regional Hospital, the Emergency and Reanimation Center of Yaoundé, Bafoussam Regional Hospital, Maroua Regional Hospital, Bafia District Hospital, Kribi District Hospital, and Bertoua Regional Hospital.

### Study population

All patients presenting with motorcycle-related injuries were eligible. Motorcycle-related injuries were defined as those sustained while riding or being a passenger on a motorcycle, or as a pedestrian struck by a motorcycle. Patients were included if they presented to a participating hospital within 14 days of injury or if they were admitted, transferred, or observed in the emergency department for more than 24 hours. Patients were excluded if they were dead on arrival, or if the injury was unrelated to a motorcycle collision event.

### Data collection

Trained CTR staff based in the emergency department enrolled eligible patients and recorded data using structured questionnaires. Data entry was performed directly into Research Electronic Data Capture (REDCap), a secure, HIPAA-compliant database. Patients were followed through their hospital course until final disposition (discharge, transfer, death, or leaving against medical advice).

The analysis included variables describing:

- Demographics: age, sex, education, residence, occupation.

- Crash characteristics: road-user role (motorcycle user vs pedestrian), helmet use, alcohol use, and injury activity.

- Injury profile: injured body regions, severity using the Abbreviated Injury Severity Score (AIS).

- Clinical care: intensive care unit admission, surgical interventions, blood transfusion, and length of stay in hospital.

- Outcomes: mortality at ED and upon discharge, disability at discharge

### Data analysis

Data were cleaned and exported from REDCap to R version 4.2.1 for analysis. Continuous variables were summarized as means with standard deviations or medians with interquartile ranges, as appropriate. Categorical variables were summarized as frequencies and percentages. Bivariate comparisons between users and pedestrians were made using chi-square or Fisher's exact tests for categorical variables and Wilcoxon rank-sum tests for continuous variables. We also reported injury severity by AIS body region. Statistical significance was at $p < 0.05$.

## Results

### Socio-demographic characteristics

A total of 2,757 motorcycle crash victims were included; 2,339 (84.8%) were motorcycle users and 418 (15.2%) were pedestrians. Age distributions differed by participant type (p<0.001), with motorcycle users clustering in young working ages of 25–34 years: 729 (31.3%) while pedestrians were comparatively older, being 60 years and older: 97 (23.4%). Females were more frequent among pedestrians (158 [37.8%]) than users (397 [17.0%], p<0.001). Marital status (p<0.001), education (p<0.001), occupation (p<0.001), and residence (urban vs rural; p=0.010) also differed between motorcycle users and pedestrian groups (Table 1).

### Crash context and behaviours

Activity at time of injury varied markedly (p<0.001). Leisure accounted for the largest share overall (1,101 [46.9%]), predominating among pedestrians (279 [77.5%]), while users were more often injured during work (658 [33.1%]) or traveling (468 [23.5%]). Alcohol involvement among patients was higher in motorcycle users (258 [14.2%]) than pedestrians (23 [7.4%], p=0.001). Helmet were available and used only in 70 (3.6%) of the participants. Delay to presentation at the CTR showed no group differences (p=0.354), with 439 (40.6%) presenting within the "golden" one hour following the crash (Table 2).

### Injury characteristics

Overall, abbreviated injury severity differed by participant type (p=0.014). Multiple severe injuries were more frequent among motorcycle users (507 [21.9%]) than pedestrians (67 [16.1%]). By anatomic region, chest injury was less frequent among pedestrians (11 [2.6%]) than users (134 [5.7%], p=0.013). Differences were not detected for head/neck (p=0.102), extremities (p=0.102), face (p=0.404) or abdomen (p=0.98) (Table 3).

**Table 1. Socio-demographic characteristics of study participants.**

| Variable | Category | Motorcycle user No (%) | Pedestrian No (%) | Total No (%) | p-value |
|---|---|---|---|---|---|
| Age group (years) | 0-14 | 57 (2.4) | 100 (24.1) | 157 (5.7) | <0.001 |
| | 15-24 | 648 (27.8) | 50 (12.0) | 698 (25.4) | |
| | 25-34 | 729 (31.3) | 53 (12.8) | 782 (28.5) | |
| | 35-44 | 505 (21.7) | 44 (10.6) | 549 (20.0) | |
| | 45-59 | 284 (12.2) | 71 (17.1) | 355 (12.9) | |
| | ≥60 | 107 (4.6) | 97 (23.4) | 204 (7.4) | |
| Sex | Female | 397 (17.0) | 158 (37.8) | 555 (20.2) | <0.001 |
| | Male | 1937 (83.0) | 260 (62.2) | 2197 (79.8) | |
| Marital status | Married/Partnered | 1030 (45.5) | 146 (36.2) | 1176 (44.1) | <0.001 |
| | Previously married | 60 (2.7) | 32 (7.9) | 92 (3.4) | |
| | Single | 1174 (51.9) | 225 (55.8) | 1399 (52.5) | |
| Education | No formal | 157 (8.2) | 46 (14.2) | 203 (9.1) | <0.001 |
| | Primary | 431 (22.6) | 132 (40.7) | 563 (25.2) | |
| | Secondary | 1096 (57.4) | 113 (34.9) | 1209 (54.1) | |
| | University | 227 (11.9) | 33 (10.2) | 260 (11.6) | |
| Occupation group | White-collar | 510 (21.8) | 85 (20.5) | 595 (21.6) | <0.001 |
| | Manual/Farming | 1304 (55.8) | 112 (27.0) | 1416 (51.5) | |
| | Non-employed | 321 (13.7) | 159 (38.3) | 480 (17.4) | |
| | Unemployed | 202 (8.6) | 59 (14.2) | 261 (9.5) | |
| Residence | Rural | 521 (22.5) | 73 (17.9) | 594 (21.8) | 0.042 |
| | Urban | 1790 (77.5) | 335 (82.1) | 2125 (78.2) | |

*"Non-employed" includes students, retirees, and homemakers. "Unemployed" refers to individuals actively seeking employment.*

## Clinical care and resource use

Blood transfusion was uncommon (22 [0.8%], p = 0.196). Orthopedic surgery was the most common operation overall (338 [48.6%]) among those with surgery recorded, with no significant difference by participant type (p = 0.362). Head and face procedures were less frequent and did not vary by group (p = 0.828 and p = 0.404, respectively). Length of stay was typically 2–7 days (1,177 [54.5%]); prolonged stays >30 days were infrequent (88 [4.1%]), and length of stay categories did not differ by participant type (p = 0.156) (Table 4).

## Patient disposition

More than half (1,641 [59.9%]) of the patients were admitted to the ward, with similar proportions of motorcycle users (1,400 [60.2%]) and pedestrians (241 [58.4%]; p = 0.660). A smaller proportion required intensive care unit admission (170 [6.2%]) or direct transfer to the operating room (60 [2.2%]). Overall, 86 patients (3.1%) died in the emergency department. Functional status at discharge did not differ by participant type (p = 0.799). Overall, 431 (19.9%) had major disability and 77 (3.6%) died prior to discharge (Table 5).

## Discussion

This study used prospective registry data from multiple trauma centres in Cameroon to comparatively characterize the epidemiology, injury characteristics, clinical care, and outcomes of motorcycle-related injuries among motorcycle users and pedestrians. Most patients were motorcycle users, concentrated among the economically active population, while

**Table 2. Crash context and behaviours of study participants.**

| Variable | Category | Motorcycle users No (%) | Pedestrian No (%) | Total No (%) | p-value |
|---|---|---|---|---|---|
| Injury activity | Work | 658 (33.1) | 46 (12.8) | 704 (30.0) | <0.001 |
| | Education | 11 (0.6) | 8 (2.2) | 19 (0.8) | |
| | Sports | 3 (0.2) | 2 (0.6) | 5 (0.2) | |
| | Leisure | 822 (41.3) | 279 (77.5) | 1101 (46.9) | |
| | Travelling | 468 (23.5) | 19 (5.3) | 487 (20.7) | |
| | Domestic activity | 27 (1.4) | 6 (1.7) | 33 (1.4) | |
| Helmet use | Available but not used | 406 (20.9) | 3 (13.0) | 409 (20.9) | 0.646 |
| | Available and used | 69 (3.6) | 1 (4.3) | 70 (3.6) | |
| | Not available | 1463 (75.5) | 19 (82.6) | 1482 (75.6) | |
| Alcohol use among patients | Yes | 258 (14.2) | 23 (7.4) | 281 (13.2) | 0.001 |
| | No | 1553 (85.8) | 288 (92.6) | 1841 (86.8) | |
| Alcohol use among motorcycle drivers | Yes | 113 (17.0) | 11 (12.8) | 124 (16.6) | 0.399 |
| | No | 550 (83.0) | 75 (87.2) | 625 (83.4) | |
| Delay to presentation at CTR | <1 hour | 364 (40.2) | 75 (42.6) | 439 (40.6) | 0.354 |
| | 1-5 hours | 276 (30.5) | 53 (30.1) | 329 (30.4) | |
| | 6-23 hours | 31 (3.4) | 10 (5.7) | 41 (3.8) | |
| | >=24 hours | 234 (25.9) | 38 (21.6) | 272 (25.2) | |

Domestic activities refer to travel or errands related to household responsibilities, including market visits, family transport, or related household tasks.

**Table 3. Injury characteristics among the study participants.**

| Variable | Category | Motorcycle user No (%) | Pedestrian No (%) | Total No (%) | p-value |
|---|---|---|---|---|---|
| Abbreviated injury severity score | None | 749 (32.3) | 132 (31.8) | 881 (32.3) | 0.014 |
| | One severe | 1060 (45.8) | 216 (52.0) | 1276 (46.7) | |
| | Multiple severe | 507 (21.9) | 67 (16.1) | 574 (21.0) | |
| Anatomic region injured | Face | 55 (67.1) | 4 (100.0) | 59 (68.6) | 0.404 |
| | Head/neck | 752 (32.2) | 152 (36.4) | 904 (32.8) | 0.102 |
| | Chest | 134 (5.7) | 11 (2.6) | 145 (5.3) | 0.013 |
| | Abdomen | 200 (8.6) | 35 (8.4) | 235 (8.5) | 0.980 |
| | Extremities | 1065 (45.5) | 209 (50.0) | 1274 (46.2) | 0.102 |

pedestrians were older and more often female. Pre-crash behaviors differed by participant type, with leisure activities predominating among pedestrians while work- or travel-related activities were more common among motorcycle users. Alcohol involvement was higher among motorcycle users. Overall injury severity was substantial, with nearly half sustaining at least one severe injury. Clinical care indicators reflected high resource needs, as more than half required admission and nearly one in ten required critical or operative care. Despite this burden, emergency mortality was low and discharge outcomes were broadly similar between groups, with approximately one in five discharged with major disability.

The observed age and sex distribution characterized by predominance of young male motorcycle users and a substantial burden among older female pedestrians mirrors prior Cameroonian reports of road-traffic injury epidemiology emphasizing both the concentration of injuries among working-age males and the vulnerability of pedestrians in urban settings [10]. The high share of disability at discharge aligns with a prior population-based survey in the South West Region of Cameroon documenting substantial functional limitations among injured persons in the community [18].

**Table 4. Clinical care and resource use among participants.**

| Variable | Category | Motorcycle user No (%) | Pedestrian No (%) | Total No (%) | p-value |
|---|---|---|---|---|---|
| Blood transfusion | Yes | 16 (0.7) | 6 (1.4) | 22 (0.8) | 0.196 |
| | No | 2323 (99.3) | 412 (98.6) | 2735 (99.2) | |
| Type of surgery performed | Face | 55 (67.1) | 4 (100.0) | 59 (68.6) | 0.404 |
| | Head | 28 (54.9) | 4 (44.4) | 32 (53.3) | 0.828 |
| | Chest | 9 (90.0) | 1 (50.0) | 10 (83.3) | 0.729 |
| | Orthopedic | 289 (49.4) | 49 (44.1) | 338 (48.6) | 0.362 |
| Length of hospital stay (days) | 0-1 | 203 (11.1) | 38 (11.8) | 241 (11.2) | |
| | 2-7 | 993 (54.1) | 184 (57.0) | 1177 (54.5) | 0.156 |
| | 8-30 | 559 (30.4) | 95 (29.4) | 654 (30.3) | |
| | >30 | 82 (4.5) | 6 (1.9) | 88 (4.1) | |

**Table 5. Patient disposition at the emergency department and at discharge.**

| Variable | Category | Motorcycle users No (%) | Pedestrian No (%) | Total No (%) | p-value |
|---|---|---|---|---|---|
| Disposition at the Emergency Department | Discharged home to die | 2 (0.1) | 0 (0.0) | 2 (0.1) | 0.660 |
| | Emergency ward observation | 243 (10.4) | 53 (12.8) | 296 (10.8) | |
| | Admitted to ward | 1400 (60.2) | 241 (58.4) | 1641 (59.9) | |
| | Admitted to ICU | 144 (6.2) | 26 (6.3) | 170 (6.2) | |
| | Directly to OR | 50 (2.1) | 10 (2.4) | 60 (2.2) | |
| | Died | 69 (3.0) | 17 (4.1) | 86 (3.1) | |
| | Left against medical advice | 353 (15.2) | 57 (13.8) | 410 (15.0) | |
| | Transferred | 66 (2.8) | 9 (2.2) | 75 (2.7) | |
| Disposition at discharge | Little or no disability | 587 (31.9) | 99 (30.5) | 686 (31.7) | 0.799 |
| | Minor disability | 817 (44.4) | 154 (47.4) | 971 (44.8) | |
| | Major disability | 370 (20.1) | 61 (18.8) | 431 (19.9) | |
| | Deceased | 66 (3.6) | 11 (3.4) | 77 (3.6) | |

ICU: Intensive Care Unit; OR: Operating Room.

The non-use of helmets and alcohol involvement are key modifiable risk factors that significantly increase the severity and fatality of road traffic injuries. Helmet non-use exposes motorcycle users to direct head trauma during crashes, while alcohol impairs judgment, reaction time, and motor coordination, elevating the risk of both crash occurrence and severity. African syntheses show that helmet use markedly reduces severe head injury and mortality [19]. Cameroon's national helmet law applies to drivers and passengers but lacks a fastening requirement, an implementation gap that may blunt protective effects and should be addressed in policy and enforcement updates [3]. Strengthened legislation and enforcement could therefore yield immediate public health gains.

Alcohol consumption among users of motorcycle further compounds injury risk. Alcohol consumption among injured motorcycle users in this study parallels earlier findings from the CTR, which reported that alcohol use before injury was independently associated with mortality among trauma patients in Cameroon [20]. This reflects a pervasive culture of unregulated drink-driving and limited roadside enforcement. There is an absence of systematic alcohol screening post-crash, which likely leads to underestimation of true exposure. Together with our findings, these results highlight the urgent

need for comprehensive drink-driving policies combining enforcement, public awareness, and post-crash testing into the national road safety strategy.

The injury severity and resource needs observed in this study likely contribute to adverse outcomes among both motorcycle users and pedestrians due to poorly organized prehospital trauma systems. LMIC reviews consistently describe fragmented prehospital systems and delays spanning care seeking, reaching definitive care, and receiving timely care [21]. Recent CTR analyses of RTI in Cameroon similarly report non-trivial in-hospital mortality and highlight opportunities to strengthen early care pathways [16]. Together with WHO's 2023 Global Status Report underscoring driving under the influence of alcohol and helmet enforcement as regional priorities, these data support integrated prevention through helmet enforcement and alcohol deterrence, coupled with system investments in improving trauma systems.

Our finding that emergency department disposition and discharge outcomes were broadly similar between users and pedestrians despite differences in injury severity suggests that downstream care processes may attenuate initial exposure differences or, alternatively, that both groups face common system constraints. Prior Cameroonian work has described substantial barriers to timely operative and rehabilitative care post-injury, with financial and logistical barriers shaping outcomes after discharge [18].

Policy measures with near-term impact include: (1) enforcement helmet use according to best practice standards and deterrence of alcohol-impaired riding among motorcycle users; (2) investment in pedestrian-safe infrastructure; and (3) investment in trauma system strengthening particularly in prehospital lay first responder training and transportation of injured victims to trauma capable facilities to reduce time-to-care [3]. In hospitals, prioritizing rapid head injury evaluation and orthopaedic capacity can better match the observed clinical demand profile in motorcycle-related crashes [19]. Given the high disability burden at discharge, linkage to follow-up and community-based rehabilitation should be integrated into trauma quality-improvement initiatives in Cameroon [18].

Strengths include prospective, registry-based capture at high-volume centres and standardized definitions (including AIS-based severity). We also report both process (disposition, surgery, transfusion, intensive care) and outcome (disability, mortality) measures, enabling system-level insights. Limitations include potential selection bias to participating facilities, missingness in some prehospital/context variables, and residual confounding in group comparisons. Helmet exposure quality (fit/fastening/standards) could not be fully characterized from routine fields, which may underestimate the "true" protective effect of high-quality, properly fastened helmets.

Future research should evaluate the effectiveness of enforcing the use of helmet, quantify financial barriers to trauma care, and assess long-term functional recovery. Building trauma system capacity through coordinated policy, financing, and quality-improvement initiatives as advocated by Juillard and colleagues [22] and the Lancet Commission on Global Surgery [23] might be essential to reduce the burden of preventable death and disability following motorcycle-related crashes in Cameroon.

## Conclusion

Motorcycle-related injuries in Cameroon disproportionately affect young, economically active males, while older female pedestrians also experience substantial injury burden. Together, these populations impose high clinical and functional demands on the health system. Strengthening primary prevention through helmet enforcement and alcohol risk reduction for motorcycle users, alongside pedestrian-focused safety and infrastructure interventions, combined with trauma system strengthening within a national framework, could contribute to reducing disability and mortality following motorcycle-related crashes in Cameroon.

## Supporting information

**S1 File. CTR dataset for Epidemiology of Motorcycle-related Injuries.**
(CSV)

## Acknowledgments

We are grateful to all individuals and institutions involved in the design, implementation, and ongoing operation of the Cameroon Trauma Registry. We thank the dedicated CTR staff, hospital administrators, and data collectors at participating hospitals for their invaluable contributions. We also acknowledge the support of the Cameroonian Ministry of Public Health, the University of Buea, and the National Institutes of Health. Finally, we extend our sincere appreciation to all trauma patients and their families whose participation made this study possible.

## Author contributions

**Conceptualization:** Elvis A. Tanue, Sandra I. McCoy.

**Data curation:** Odette D. Kibu, Alan Hubbard, Arole Darwin Touko, Fanny Nadia Dissak-Delon, Alain Chichom-Mefire.

**Formal analysis:** Elvis A. Tanue, Betrand A. Tambe.

**Funding acquisition:** Alan Hubbard, Sandra I. McCoy, Catherine Juillard, Alain Chichom-Mefire.

**Investigation:** Elvis A. Tanue, Nicholas Tendongfor, Alan Hubbard, Nahyeni Bassah, Rasheedat Oke, Arole Darwin Touko, S. Ariane Christie, Georges Nguefack-Tsague, Dickson S. Nsagha, Alain Chichom-Mefire.

**Methodology:** Odette D. Kibu, Betrand A. Tambe, Alan Hubbard, Isaac Obeng-Gyasi, Arole Darwin Touko, Fanny Nadia Dissak-Delon, S. Ariane Christie, Georges Nguefack-Tsague, Dickson S. Nsagha, Catherine Juillard, Alain Chichom-Mefire.

**Project administration:** Nicholas Tendongfor, Alan Hubbard, Sandra I. McCoy, Rasheedat Oke, Fanny Nadia Dissak-Delon, Georges Nguefack-Tsague, Catherine Juillard, Alain Chichom-Mefire.

**Resources:** Sandra I. McCoy.

**Supervision:** Nicholas Tendongfor, Alan Hubbard, Sandra I. McCoy, Nahyeni Bassah, Rasheedat Oke, Isaac Obeng-Gyasi, Fanny Nadia Dissak-Delon, S. Ariane Christie, Georges Nguefack-Tsague, Dickson S. Nsagha, Catherine Juillard, Alain Chichom-Mefire.

**Validation:** Elvis A. Tanue, Betrand A. Tambe.

**Visualization:** Elvis A. Tanue.

**Writing – original draft:** Elvis A. Tanue.

**Writing – review & editing:** Elvis A. Tanue, Odette D. Kibu, Betrand A. Tambe, Nicholas Tendongfor, Alan Hubbard, Sandra I. McCoy, Nahyeni Bassah, Rasheedat Oke, Isaac Obeng-Gyasi, Arole Darwin Touko, S. Ariane Christie, Georges Nguefack-Tsague, Dickson S. Nsagha, Catherine Juillard, Alain Chichom-Mefire.

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
