## [Decision Letter · Decision Letter 0]

25 Feb 2026

PGPH-D-25-04054

Epidemiological Patterns of Motorcycle-Related Injuries in Cameroon: A Comparative Analysis of Motorcycle Users and Pedestrians

Dear Dr. Tanue,

Thank you for submitting your manuscript to PLOS Global Public Health. After careful consideration, we feel that it has merit but does not fully meet PLOS Global Public Health’s publication criteria as it currently stands. Therefore, we invite you to submit a revised version of the manuscript that addresses the points raised during the review process.

**Please respond to the minor revisions requested by the reviewer, focusing on clarifications of categories in the tables and using less stigmatizing language in parts. (e.g., consider changing "perpetrator" to "motorcycle driver" )**

We look forward to receiving your revised manuscript.

Kind regards,

Hani Mowafi, M.D., M.P.H.

Academic Editor

Journal Requirements:

1. Please ensure that your Ethics Statement is available in its entirety at the beginning of your Methods section, under a subheading 'Ethics Statement'.

2. We note that your Data Availability Statement is currently as follows: All relevant data are within the manuscript.

Additional Editor Comments (if provided):

Please responsd to some of the clarifications requested by Reviewer. Specifically, clarify the categories in the tables and change "perpetrator" to motorcycle driver

Reviewers' comments:

Reviewer's Responses to Questions

**Comments to the Author**

1. Does this manuscript meet PLOS Global Public Health’s publication criteria? Is the manuscript technically sound, and do the data support the conclusions? The manuscript must describe methodologically and ethically rigorous research with conclusions that are appropriately drawn based on the data presented.? Is the manuscript technically sound, and do the data support the conclusions? The manuscript must describe methodologically and ethically rigorous research with conclusions that are appropriately drawn based on the data presented.

Reviewer #1: Yes

Reviewer #2: Yes

2. Has the statistical analysis been performed appropriately and rigorously?

Reviewer #1: Yes

Reviewer #2: Yes

3. Have the authors made all data underlying the findings in their manuscript fully available (please refer to the Data Availability Statement at the start of the manuscript PDF file)?

The PLOS Data policy requires authors to make all data underlying the findings described in their manuscript fully available without restriction, with rare exception. The data should be provided as part of the manuscript or its supporting information, or deposited to a public repository. For example, in addition to summary statistics, the data points behind means, medians and variance measures should be available. If there are restrictions on publicly sharing data—e.g. participant privacy or use of data from a third party—those must be specified.requires authors to make all data underlying the findings described in their manuscript fully available without restriction, with rare exception. The data should be provided as part of the manuscript or its supporting information, or deposited to a public repository. For example, in addition to summary statistics, the data points behind means, medians and variance measures should be available. If there are restrictions on publicly sharing data—e.g. participant privacy or use of data from a third party—those must be specified.

Reviewer #1: Yes

Reviewer #2: No

4. Is the manuscript presented in an intelligible fashion and written in standard English?

Reviewer #1: Yes

Reviewer #2: Yes

Reviewer #1: This manuscript presents a comparative analysis of motorcycle-related injuries among motorcycle users and pedestrians using data from the Cameroon Trauma Registry. While the study does not introduce a novel methodological approach or previously unreported risk factors, it provides multicentre, prospectively collected data from a low- and middle-income country where such evidence remains limited. In that sense, it represents a useful and necessary contribution to the literature.

The strengths of the study lie in its relatively large sample size, inclusion of multiple hospitals across different regions, standardized data collection through a trauma registry, and clear stratification of outcomes between motorcycle users and pedestrians. The descriptive epidemiology is sound and confirms patterns reported from other studies from LMICs - low helmet use, alcohol involvement, high hospitalization rates, and substantial post-injury disability. Importantly, the manuscript highlights the often-underreported burden among pedestrians, particularly older women, which adds contextual value for policy discussions.

That said, the findings are largely confirmatory rather than novel. The discussion occasionally reiterates well-established recommendations (helmet enforcement, alcohol deterrence, trauma system strengthening) without offering substantially new insights into implementation or prioritization.

Overall, despite limited novelty, this manuscript provides much needed epidemiological data from an LMIC setting and fills an important evidence gap. I believe it is suitable for publication, as it adds a valuable data point to the global road injury literature and can inform context-specific prevention and trauma system planning.

Recommendation: Accept.

Reviewer #2: Thank you for this well done job.

Only a few minor comments arise.

1) The sequence of reporting of the ethical approval presents a subtle focus on the California ethical approval for a Cameroonian registry research. Authors should consider flipping this- both Cameroonian approvals should be placed first in sequence, and then California approvals.

2) Table 1 conflates categories which should be clarified.

i) Divorced can be single or living with partner.

ii) Married can be living with partner.

iii) Widowed can be single or living with partner.

Consider collapsing categories as appropriate. Footnotes would be helpful for clarity

iv) Non-employed versus unemployed? Footnotes would be helpful for clarity

3) Clarify what "domestic activities" lead to motorcycle crashes, perhaps with a footnote to the table with examples. This is not intuitive that domestic activities in the house would lead to motorcycle crashes.

4) The term perpetrator in Table 2 (“Alcohol use among perpetuator”) is almost vilifying. Can this be replaced? Again if perpetuator means the rider who “crashed” on the pedestrian or pinion/passenger, can this individual not also suffer injury and be a patient?

An finally,

5) So that tables can be standalone, please ensure that abbreviations are embedded in them.

All underlying data are not within the manuscript- results are included in some places, but not underlying data. For instance- for Abbreviated injury severity score- we do not have the data of the scores, but of the category. For several others, we have counts and percentages, but not underlying data that we can use to cross check analyses. If there are caveats with sharing registry data, kindly note them rather than stating that all data is within the manuscript.

Thank you.

**Do you want your identity to be public for this peer review?** For information about this choice, including consent withdrawal, please see our Privacy Policy..

Reviewer #1: **Yes:**Professor Dhananjaya SharmaProfessor Dhananjaya SharmaProfessor Dhananjaya SharmaProfessor Dhananjaya Sharma

Reviewer #2: No

---

## [Editor Report · Decision Letter 1]

30 Mar 2026

Epidemiological Patterns of Motorcycle-Related Injuries in Cameroon: A Comparative Analysis of Motorcycle Users and Pedestrians

PGPH-D-25-04054R1

Dear Dr. Tanue,

We are pleased to inform you that your manuscript 'Epidemiological Patterns of Motorcycle-Related Injuries in Cameroon: A Comparative Analysis of Motorcycle Users and Pedestrians' has been provisionally accepted for publication in PLOS Global Public Health.

Best regards,

Hani Mowafi, M.D., M.P.H.

Academic Editor